

# Predictive modelling of the distribution of *Clematis* sect. *Fruticella s. str.* under climate change reveals a range expansion during the Last Glacial Maximum

Mingyu Li[*], Jian He[*], Zhe Zhao, Rudan Lyu, Min Yao, Jin Cheng and Lei Xie

Beijing Forestry University, Beijing, China
[*] These authors contributed equally to this work.

## ABSTRACT

**Background**. The knowledge of distributional dynamics of living organisms is a prerequisite for protecting biodiversity and for the sustainable use of biotic resources. *Clematis* sect. *Fruticella s. str.* is a small group of shrubby, yellow-flowered species distributed mainly in arid and semi-arid areas of China. Plants in this section are both horticulturally and ecologically important.

**Methods**. Using past, present, and future environmental variables and data with Maximum Entropy (Maxent) modeling, we evaluated the importance of the environmental variables on the section's estimated distributions, thus simulating its distributional dynamics over time. The contractions and expansions of suitable habitat between the past and future scenarios and the present were then compared.

**Results and Discussion**. The models revealed that the areas with high and moderate suitability currently encompass about 725,110 km$^2$. The distribution centroid location varies between points in Ningxia and Inner Mongolia during the different scenarios. Elevation, Mean UV-B of Lowest Month, Precipitation of Coldest Quarter, and Mean Temperature of Driest Quarter were major factors determining the section's distribution. Our modeling indicated that *Clematis* sect. *Fruticella* underwent a significant range contraction during the last interglacial period, and then expanded during the last glacial maximum (LGM) to amounts like those of the present. Cold, dry, and relatively stable climate, as well as steppe or desert steppe environments may have facilitated range expansion of this cold-adapted, drought-resistant plant taxon during the LGM. Predicted future scenarios show little change in the amounts of suitable habitat for *Clematis* sect. *Fruticella*. This study aids understanding of the distributional dynamics of *Clematis* sect. *Fruticella*, and the results will help the conservation and sustainable use of these important woody plants in Chinese arid and semiarid areas.

Corresponding authors
Jin Cheng, chengjin@bjfu.edu.cn
Lei Xie, xielei@bjfu.edu.cn

## INTRODUCTION

Ecological habitats and distributions of living organisms, community compositions, ecosystem structures, and global biodiversity have been significantly influenced by climate changes during the Quaternary period (*Qiu, Fu & Comes, 2011*; *Doxford & Freckleton, 2012*; *Pio et al., 2014*; *Carvalho et al., 2015*; *Sun et al., 2015*; *Matías et al., 2017*; *Qin et al., 2017*; *Zhang et al., 2018a*; *Zhang et al., 2018b*; *Wang et al., 2019a*; *Wang et al., 2019b*; *Mohammadi et al., 2019*). During the last glaciation, global temperature was 5–12 °C lower than they are now and the glacier areas were 8.4 times of the present time in China (*Wang & Liu, 2001*; *Li et al., 2004*; *Chen, Kang & Liu, 2011*). This greatly affected plant distributions and many plant species significantly shrank their habitat during that time (*Xu et al., 2017*; *Li et al., 2019*). Furthermore, the ongoing changing climate of the past few decades has also been affecting living systems and the geographic distributions of organisms (*Parmesan & Yohe, 2003*), and resolving the relationships between vegetation and climate dynamics has been a critical issue of ecology study (*Bellard et al., 2012*). The Fifth Assessment Report of the Intergovernmental Panel on Climate Change (IPCC) estimated that by the end of the 21th century, average global temperatures will have increased by 0.3–4.5 °C as greenhouse gas emissions continue to increase (*IPCC, 2013*). This will have major effects on future ecosystems and species distributions (*Walck et al., 2011*; *Wang et al., 2019a*; *Wang et al., 2019b*; *Li et al., 2019*). The distribution ranges of many plant species in the Northern Hemisphere may shift northward or to higher elevations in response to the future climate changes (*Hof et al., 2011*; *Bai, Wei & Li, 2018*; *Wang et al., 2019a*; *Wang et al., 2019b*; *Zhang, Zhang & Tao, 2019*).

Modeling the effects of climate change on the past and present habitats of different species across a landscape aids in understanding the organisms' potential responses to their changing environments going forward (*Wang et al., 2019a*; *Wang et al., 2019b*; *Li et al., 2019*; *Zhang, Zhang & Tao, 2019*). Species distribution modeling (SDM) is widely used and has become a valuable tool for predicting species habitat suitability (*Barbet-Massin et al., 2012*; *Miller, 2012*; *Zhang et al., 2012*; *Li & Wang, 2013*). With the availability of massive digitalized specimen records and high-resolution environmental data, SDM is widely used to assess current potential geographical habitats and evaluate the effects of climate change in recent years (*Konowalik, Procków & Procków, 2017*; *Wang et al., 2018*; *Li et al., 2019*). To accomplish such analyses, many algorithms have been developed in the past decades, such as DOMAIN (*Carpenter, Gillison & Winter, 1993*), Maximum Entropy algorithm (Maxent) (*Phillips, Anderson & Schapire, 2006*), Generalized Linear Model (GLM) (*Hirzel, Helfer & Metral, 2001*), and the Generalized Additive Model (GAM) (*Austin et al., 2009*). Of all these models, Maxent has been a powerful tool and commonly used to assess the impacts of environmental factors on the suitable habitat of a given taxon (*Hernandez et al., 2006*; *Bai, Wei & Li, 2018*; *Li et al., 2019*).

Drylands, including arid, semi-arid, and dry sub-humid ecosystems, constitute one of the earth's largest terrestrial biomes. They cover 41% of the earth's surface and support 38% of the global human population (*Maestre et al., 2012*). In China, drylands cover approximately 50% of the total land surface, and desertification has increased rapidly in

recent years. A recent study discovered that semi-arid regions that dominate the drylands in northern China have experienced much warming and significant expansion over the last 60 years (*Huang et al., 2019*). Because these regions (especially in the Loess Plateau of northern China) have suffered from overgrazing, subsistence farming, deforestation, and severe water loss and soil erosion, the ecosystems in these regions are highly vulnerable to effects rooted in the current global climate changes (*Jiang et al., 2013*; *Liu et al., 2013a*; *Liu et al., 2013b*). Thus, the knowledge of the distributional shifts of dryland vegetation responding to climate change in that area is urgently needed to aid conservation and sustainability of its biodiversity.

*Clematis* sect. *Fruticella s. str.* (Ranunculaceae), consisting of several erect, shrubby species that grow mainly on the Loess Plateau and in the northern Hengduan Mountains, is one of the most important plant taxa in northern China's arid and semi-arid ecosystems (*He, Liu & Xie, 2018*). Previous classifications broadly included all the shrubby *Clematis* species in this section (*Prantl, 1888*; *Handel-Mazzetti, 1939*; *Tamura, 1967*; *Tamura, 1995*; *Chang, 1980*; *Johnson, 1997*; *Grey-Wilson, 2000*). However, the most recent taxonomic revisions (*Wang, 2003*; *Wang & Li, 2005a*; *Wang & Li, 2005b*) considered that this broadly define section may not be a natural group due to its diversified floral traits. So, they re-defined this section to include only five species with yellow, bell-shaped flowers. Because of their white, spreading flowers, other, previously included, shrubby species (*C. lancifolia*, *C. delavayi*, *C. ispahanica*, *C. songarica*, and *C. phlebantha*), were excluded (*Wang, 2003*; *Wang & Li, 2005b*). Although the recent molecular phylogenetic studies did not include all of the *Clematis* sect. *Fruticella* member species, yellow-flowered *C. fruticosa* was found to be distantly related to white-flowered *C. delavayi* and *C. ispahanica* (*Lehtonen, Christenhusz & Falck, 2016*), thus supporting the exclusion of the white-flowered species from the section.

According to the current taxonomic revision, *Clematis* sect. *Fruticella s. str.* (hereinafter inclusive) contains *C. fruticosa*, *C. tomentella*, *C. canescens*, *C. nannophylla*, and *C. viridis* (*Wang, 2003*; *Wang & Li, 2005a*; *Wang & Li, 2005b*). The former four species are found in the vast arid and semi-arid areas of the Loess Plateau of northern China, and some populations extend to the Gobi of Mongolia. The latter species, *C. viridis*, grows on the windy and arid slopes of north Hengduan Mountains (Fig. 1) and the uppermost Yangtze River's valley. Unlike most of the other climbing species of *Clematis*, sect. *Fruticella* has strikingly different characteristics. Besides being erect shrubs, the species' well-branched woody stems often have simple and reduced leaves, specialized adaptations that minimize water loss in their arid and semi-arid habitats. Plants of *Clematis* sect. *Fruticella* are cold and drought resistant and heliophilous. The pollination system of this section is obligate out-crossing but self-compatible. The major pollinators are *Amegill aparhypate* and *Apis cerana* (*Hou et al., 2016*).

Plants in *Clematis* sect. *Fruticella* play important roles in the dryland ecosystems in northern China and have great horticultural value. *Clematis fruticosa* and *C. nannophylla* are typical vegetation components of Chinese Loess Plateau and are often used in raised bed and rock garden landscapes (*Grey-Wilson, 2000*). Because of its drought resistance and easy propagation, *C. tomentella*, along with *Hedysarum scoparium*, *Caragana korshinskii*, *Calligonum mongolicum*, and other xerophytic species, was used in straw checkerboard

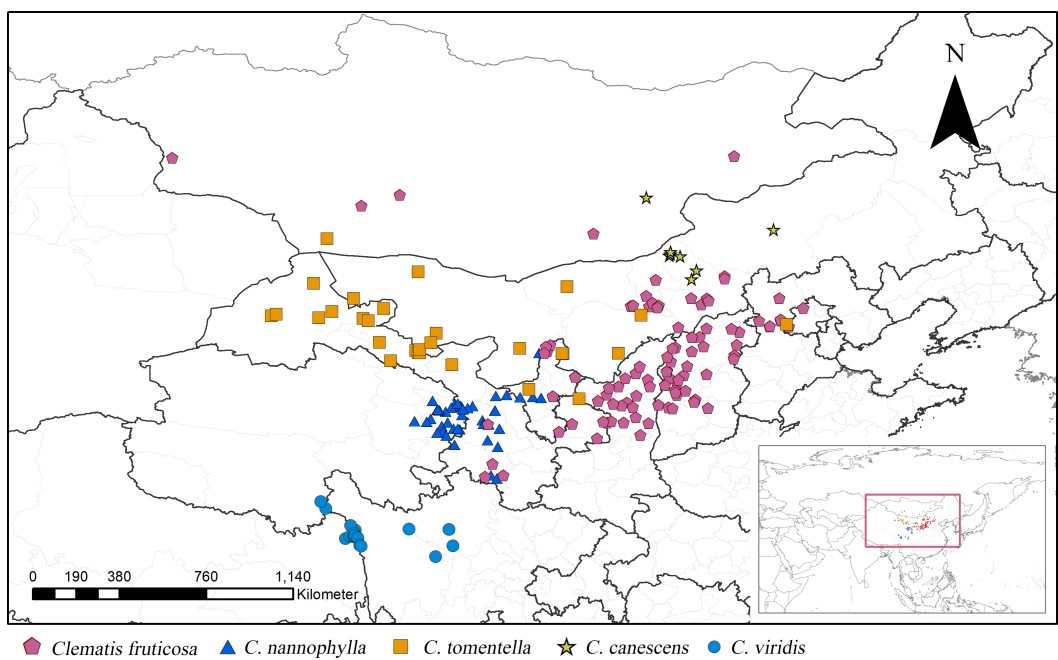

**Figure 1** Distribution of *Clematis* sect. *Fruticella* in China and Mongolia.

barriers constructed to stabilize sand in the National Nature Reserves in Ningxia Province (*Guo et al., 2012*; *Ding et al., 2013*). With its excellent stress tolerance, *Clematis fruticosa* would also be a good candidate for sand fixing and afforestation (*Liu et al., 2016*).

However, previous studies have focused mainly on the drought-resistance and horticultural utility of this section (*Guo et al., 2012*; *Ding et al., 2013*; *Liu et al., 2016*). Few studies have examined the habitat and ecological factors that contribute to its past, current, and future distribution under climate changes. Expansion and contraction of its range during Quaternary glaciations may provide a forecast of future distributional fluctuations under climate change and show key habitat suitability factors that are crucial for maintaining its utility and for management planning. In this study, we propose hypotheses that during the Last Glacial Maximum (LGM) the distribution area of *Clematis* sect. *Fruticella* was the smaller than that of the current, and in the future the distribution centroid of this section will move northward or higher elevation in response to global warming.

The ecological niche modeling (ENM) analysis is typically performed for single species. Only a few studies performed ENM analysis at higher taxonomic levels, e.g., for the willow family (Salicaceae) as a whole (*Li et al., 2019*). The present study used ENM to simulate the potential distribution of *Clematis* sect. *Fruticella* for the following reasons: (1) The five species are closely related and share a similar climatic niche on a larger spatial scale. As emphasized by *Li et al. (2019)*, one premise of ENM is that the ecological requirements and distributions of species are in equilibrium (*Peterson et al., 2011*). All the members of *Clematis* sect. *Fruticella* are important vegetation components in dryland of China. The

habitat is unique in the genus *Clematis* and even in Ranunculaceae (*Wang & Li, 2005a*; *He, Liu & Xie, 2018*); (2) The accuracy of specimen identification is crucial for robust niche modeling and will greatly affect the simulation results. The five species of *Clematis* sect. *Fruticella* are so closely related and similar morphologically that the species identification within this section are problematic in all the available online databases. For example, we can obtain 125 specimen records under the name *Clematis canescens* in Chinese Virtual Herbarium (CVH, http://www.cvh.org.cn). However, more than 90% of the specimens are misidentified according to the most current taxonomic revision (*Wang & Li, 2005a*). Most of the 125 specimens are actually *C. tomentella*, some of them are *C. viridis* and *C. fruticosa*. The distribution data is likely biased by incorrectly identified species, whereas the identification at the section level is far less prone to mis-identifications. In this case, we can consider *Clematis* sect. *Fruticella* as a species aggregate. (3) We combined the occurrence data of *Clematis* sect. *Fruticella* so the ENM results were able to fully convey this section's geographical distribution patterns.

In the present study, we obtained a comprehensive occurrence dataset of *Clematis* sect. *Fruticella*, as well as high-resolution environmental data of its distribution areas in China and Mongolia. We used that data in Maxent to build species distribution models to project and quantify *Clematis* sect. *Fruticella* habitat changes under a variety of past, present, and future climate scenarios to determine what influences the distributional and ecological patterns of this group of plants. Our results offer a reference to assist in conservation and sustainable utility of shrubby *Clematis* resources for afforestation in the drylands of China.

## MATERIALS & METHODS

### Spatial data compilation

*Clematis* sect. *Fruticella* is distributed mainly in northern China, with one species (*C. viridis*) distributed in the north Hengduan Mountains and some populations of *C. fruticosa* and *C. canescens* occurring in Mongolia. In those areas, the annual precipitation of 110–750 mm is lowest in the north and increases going southward, as does the annual average temperature of −2 to 10 °C. Precipitation also decreases from the east to the west. Vegetation types are alpine shrub grassland along river valleys for *C. canescens*, and desert shrub grassland and desert grassland for the other species.

From July 2015 to May 2019, we conducted several multisite surveys for *Clematis* sect. *Fruticella* across its whole distribution area, including eastern Qinghai, northwest Sichuan, western Gansu, Ningxia, Shanxi, Inner Mongolia, and Hebei provinces. Our distribution data were then compiled from both our field data and the CVH online database. Two other online databases, the Global Biodiversity Information Facility (http://www.gbif.org; GBIF Occurrence Download, https://doi.org/10.15468/dl.nuge5a) and the National Specimen Information Infrastructure of China (http://www.nsii.org.cn), were also consulted. We carefully checked the distribution records and identified specimens according to the current taxonomic revision (*Wang & Li, 2005a*). Distribution records without geographic coordinates were assigned coordinates using Google Earth (http://www.google.com/earth). Duplicates were removed from our analyses. Eight distribution sites for *C. canescens*, 19

for *C. viridis*, 29 for *C. tomentella*, 43 for *C. nannophylla*, and 101 for *C. fruticosa* were documented. In total, 200 records of *Clematis* sect. *Fruticella* were used in our analysis (Table S1).

## Environmental variables
### Current environmental scenario

Plants of *Clematis* sect. *Fruticella* are cold and drought resistant and heliophilous. The growth and reproduction of this section may have been determined by various interacting factors, such as temperature, precipitation, solar radiation, and soil (*Zhen & Liu, 2010*; *Liu et al., 2014*; *Yu et al., 2018*; *He, Liu & Xie, 2018*). Therefore, in this study, 35 environmental variables, including climate, soil, UV-B radiation, and topography, were chosen for niche modeling analysis. We obtained 19 bioclimatic variables with 2.5′ spatial resolution (also referred to as 5 km$^2$ spatial resolution) from WorldClim-Global Climate Data (http://www.worldclim.org/bioclim) (*Hijmans et al., 2005*) and applied them to current, past, and future conditions. We also downloaded four datasets (Wet days (wet), Ground frost frequency (frs), Water vapor (vap), and Cloud cover (cld)) from the IPCC (http://www.ipcc-data.org/observ/clim/cru_ts2_1.html) (*Mitchell & Jones, 2005*). Six UV variables (Annual Mean UV-B (UVB1), UV-B Seasonality (UVB2), Mean UV-B of Highest Month (UVB3), Mean UV-B of lowest Month (UVB4), Sum of Monthly Mean UV-B during Highest Quarter (UVB5), and Sum of Monthly Mean UV-B during Lowest Quarter (UVB6) were obtained from gIUV, a global UV-B radiation dataset for macroecological studies (https://www.ufz.de/gluv/index.php?en=32435) (*Beckmann et al., 2014*). Growing degree days, soil pH, and soil organic carbon were acquired from the University of Wisconsin (http://nelson.wisc.edu/sage/data-and-models/atlas/maps.php) (*New, Hulme & Jones, 1999*) and three other variables: barren/very sparsely vegetated land (NVG), mapped water bodies, and global elevation (GloElev) were downloaded from the Food and Agriculture Organization (*IIASA; FAO, 2012*). All these environmental variables were transferred into ASCII format using ArcGIS 10.2 Conversion Tools and then overlapped with map of Asia that was obtained from the Database of Global Administrative Areas (https://www.gadm.org/maps.html) and then used the result to extract environmental data.

### Multicollinearity analysis among variables

Because many environmental variables are spatially correlated and strong correlation between the environmental variables may cause over-fitting and imprecise modeling (*Hu & Liu, 2014*), we first excluded environmental variables with 0 contribution values using the Maxent Jackknife test (*Aguirre-Gutiérrez et al., 2013*). Then, we conducted a multicollinearity test in ArcGIS 10.2 to evaluate relationships between the rest of the environmental variables and to eliminate overcounting bias. Finally, we calculated Pearson correlation coefficients using the Band Collection Statistics tool in ArcGIS 10.2 to check the correlations among the 35 bioclimatic variables (*Fourcade et al., 2014*) and eliminate sampling bias. If a pair of variables were highly correlated ($r \geq 0.8$), only the most ecologically meaningful variables were kept (*Aguirre-Gutiérrez et al., 2013*). As a result

of these analyses, out of the original 35 environmental variables we retained nine for subsequent evaluation.

## The past and future climate scenarios and years

Climate changes during the late Pleistocene, especially the last glacial–interglacial cycle, greatly impacted vegetation and caused forest types to fluctuate in East Asia (*Fang, 1991*; *Zhou, Qiu & Guo, 1991*; *Harrison et al., 2001*). Because of this, we used three paleoclimate data sets: the Last Interglacial period (LIG, about 120–140 ka), the Last Glacial Maximum (LGM, about 22 ka), and the Mid Holocene (MH, about 6 ka) in the Community Climate System Model four (CCSM4) global climate model (http://www.worldclim.org) to predict possible past distribution areas. CCSM4 is one of the most efficient climate models that can simulate the influence of past and future climatic changes on the distribution of plants (*Abdelaal et al., 2019*). This model has been already successfully applied in similar studies (*Arar et al., 2019*; *Li, Fan & He, 2020*).

We applied the Representative Concentration Pathways (RCPs) defined in the IPCC Fifth Assessment Report to predict future conditions. These models better interpret how climate processes work than do the models in the previous four assessment reports (*IPCC, 2013*). RCPs, including RCP 2.6, RCP4.5, RCP6.0, and RCP 8.5, are potential pathways of radiative forcing values (measured as +2.6, +4.5, +6.0, and +8.5 W/m$^2$, respectively) in the year 2100 relative to pre-industrial values. We used four RCP combinations with the CCSM4 climate change modeling data, where 2050 and 2070 use average emissions for the years 2041 to 2060 and 2061 to 2080, respectively. For modeling analyses, we used the RCP 2.6–2050, RCP 2.6–2070, RCP 8.5–2050, and RCP 8.5–2070 combinations with CCSM4 to simulate global climate responses to increased greenhouse gas emissions.

## Model simulation

Based on distributional data and the selected environmental parameters, we used Maxent for SDM of past, current, and future climate conditions for the whole section, as well as for each species (*Phillips et al., 2017*). To estimate the capacity of the model, 25% of the data was used for testing, while 75% of the location point data was used for training. SDM was performed using the factors discussed in the previous paragraph, along with occurrence data. The algorithm either ran 1,000 iterations of the processes or continued until convergence was reached (threshold, 0.00001). Two methods were utilized to evaluate Maxent performance: the area under the receiver-operating characteristic curve (AUC) (*Fielding & Bell, 1997*; *Babar et al., 2012*) and true skill statistic (TSS) (*Allouche, Tsoar & Kadmon, 2006*). AUC is a threshold-independent measurement for assessing model performance (*Fielding & Bell, 1997*). The value of AUC varies from 0 to 1, with AUC more than 0.9 thought to be good prediction (*Swets, 1988*). Whereas, TSS varies from −1 and +1 and takes both omission and commission errors into account. TSS value from 0.70–0.85 indicates good performance, from 0.85–1.00 indicates excellent performance (*Luo, Wang & Lyu, 2017*). The R package Biomod2 was applied to conduct TSS assessment (*Thuiller et al., 2009*; *Thuiller et al., 2016*).

We used the jackknife test to assess the significance of those variables, obtaining a range of values, which we reclassified into four categories of potential suitable habitat: high

(>0.6), moderate (0.4–0.6), low (0.2–0.4), and none (<0.2) in the final potential species distribution map (*Zhang, Zhang & Tao, 2019*). Maxent generated a current, three past, and four future climate scenarios to calculate the SDM projections. The distribution shifts between two scenarios were explained by the movement of the central point (the centroid) of the suitable habitat. Suitable habitat changes from one period to the next were estimated by cross-checking the suitable habitat areas in the past and future scenarios against the present distribution area.

## RESULTS

### Model accuracy and suitable areas for *clematis* sect. *Fruticella*

Models for *Clematis* sect. *Fruticella* with a cross-validation AUC of 0.986 and a TSS value of 0.901 (excellent performance) indicated that the Maxent model can accurately predict the location of potential suitable habitat (Fig. 2). The potential suitable habitat for the current scenario included Hebei, Inner Mongolia, Shanxi, Shaanxi, Ningxia, Gansu, Qinghai, Sichuan, Xizang, Xinjiang, Yunnan, and Liaoning provinces of China and Mongolia (Fig. 2D). Among those, the most suitable areas are concentrated in the Loess Plateau and adjacent areas of northern China. The highly suitable areas cover 238,605 km$^2$, the moderately suitable areas covers 486,505 km$^2$, and the least suitable areas encompass 665,699 km$^2$. The areas with low, medium, and high suitable habitats in the past and future scenarios are in Table S2.

### Key environmental factors influencing the current habitat

Jackknife results showed that among the proportions of variation of the nine model variables, GloElev (22.9% of variation), UVB4 (22.2%), Precipitation of Coldest Quarter (BIO19, 20.7%), and Mean Temperature of Driest Quarter (BIO9, 18.5%) contributed the greatest weights for the current scenario (Fig. 3, Table 1). The cumulative contributions of these four environmental variables (84.3%) were also the most important variables with the highest weights in the seven past and future scenarios (Table S3).

Based on the major variables' response curves, we obtained the thresholds (existence probability >0.2): GloElev, ranging from 539 to 3,620 m; UV-B4, more than 300.5 J/m$^2$/day; all the tested amounts of BIO19, with a peak from 0–47.6 mm and; and BIO9, ranging from −20.7 to 12.3 °C (Fig. 4).

### Suitable areas under climate changes

The SDM of the LIG (Fig. 5B) estimates that southern and western Xinjiang, western and central Inner Mongolia, western Henan, northern Yunnan, southern Xizang, and Central Pakistan, had more suitable habitat (153,522 km$^2$) than the current distributions. However, the suitable habitat in Xinjiang, Gansu, Ningxia, Inner Mongolia, western Sichuan, Shaanxi, Hebei and eastern Qinghai and Xizang provinces of China decreased significantly, and the habitat in Mongolia was completely lost. In total, there was 730,700 km$^2$ less suitable habitat in the LIG than there is now. During the LGM, and compared to current amounts, estimated suitable habitat increased by 173,431 km$^2$, mainly in Gansu, Shaanxi, Shanxi, Henan, Inner Mongolia, Sichuan, and Xinjiang provinces of China, but 174,027 km$^2$ was
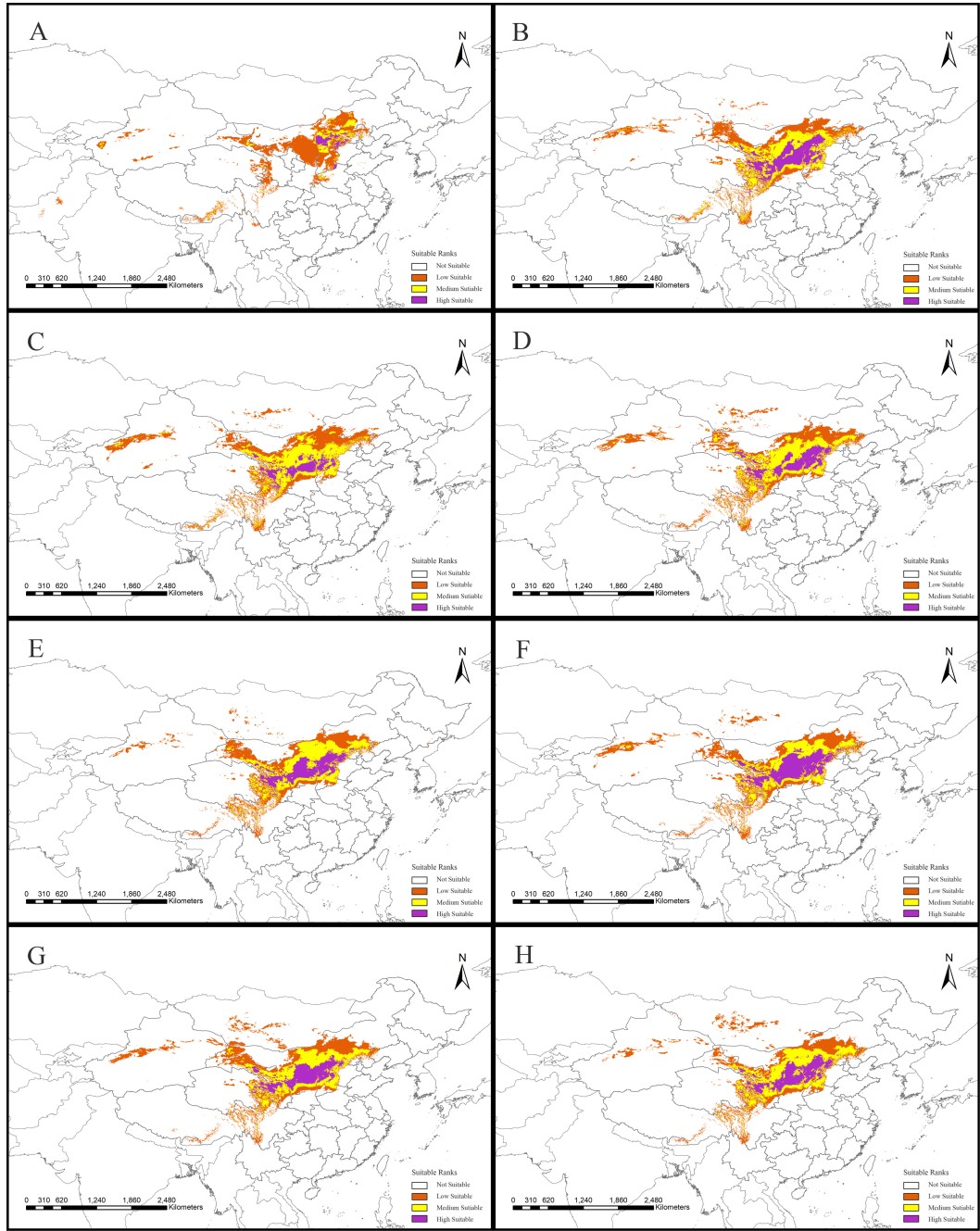

**Figure 2 Potential suitable distribution areas for *Clematis* sect. *Fruticella* predicted using Maxent modeling of eight scenarios.** (A) Last interglacial (B) Last Glacial Maximum. (C) Mid Holocene. (D) Current. (E) Year of 2050 (under RCP2.6). (F) Year of 2070 (under RCP2.6). (G) Year of 2050 (under RCP8.5). (H) Year of 2070 (under RCP8.5). RCP: representative concentration pathway.

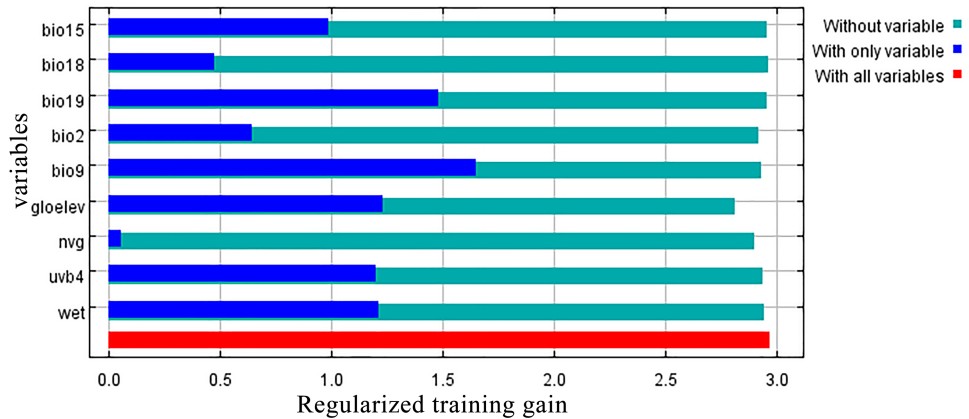

**Figure 3** Jackknife test results showing the relative importance (training gain) of the nine selected environmental variables for *Clematis* sect. *Fruticella*.

**Table 1** Percentage contributions and permutation importance of the variables included in the Maxent models for *Clematis* sect. *Fruticella*.

| Environmental variables | Resolution | Unit | Contribution (%) |
|---|---|---|---|
| Elevation (GloElev) | $30'' \times 30''$ | m | 22.9 |
| Mean UV-B of Lowest Month (UVB4) | $15' \times 15'$ | J/m$^2$/day | 22.2 |
| Precipitation of Coldest Quarter (BIO19) | $2.5' \times 2.5'$ | mm | 20.7 |
| Mean Temperature of Driest Quarter (BIO9) | $2.5' \times 2.5'$ | °C ×10 | 18.5 |
| Barren/very sparsely vegetated land (NVG) | $5' \times 5'$ | | 7.5 |
| Mean Diurnal Range (Mean of monthly (max −min) temp) (BIO2) | $2.5' \times 2.5'$ | °C ×10 | 2.8 |
| Wet days (WET) | $30' \times 30'$ | d. | 2.2 |
| Precipitation of Warmest Quarter (BIO18) | $2.5' \times 2.5'$ | mm | 2 |
| Precipitation Seasonality (Coefficient of Variation) (BIO15) | $2.5' \times 2.5'$ | mm | 1.1 |

lost, mainly in the northern, northeastern, and southwestern areas of the current suitable habitat (Fig. 5C, Table 2). The MH model was similar to that of the LGM, with estimated habitat losses of 137,774 km$^2$, mainly in the northern and northeastern distribution areas, the Hengduan Mountains of China, and in Mongolia, but with increases (147,854 km$^2$) in the northern, southern, and western parts of the current habitat.

Of the future scenarios, RCP2.6 is based on low levels of greenhouse gas emissions. Maxent estimated that the suitable habitat would increase by 109,356 km$^2$ by 2050 under that pathway, mainly in eastern Qinghai-Xizang, western Sichuan, northern Gansu, and Inner Mongolia of China and in Mongolia, but lost suitable habitat of 173,101 km$^2$ would occur mainly in Central Xinjiang, western Inner Mongolia, southern Sichuan, and northern Yunnan provinces of China and in Mongolia (Fig. 5E, Table 2). By 2070, Maxent estimated that the range of suitable habitat would increase by 124,350 km$^2$ mainly in the northern edge of the current habitat, but 132,483 km$^2$ of suitable habitat would be lost (Fig. 5F,

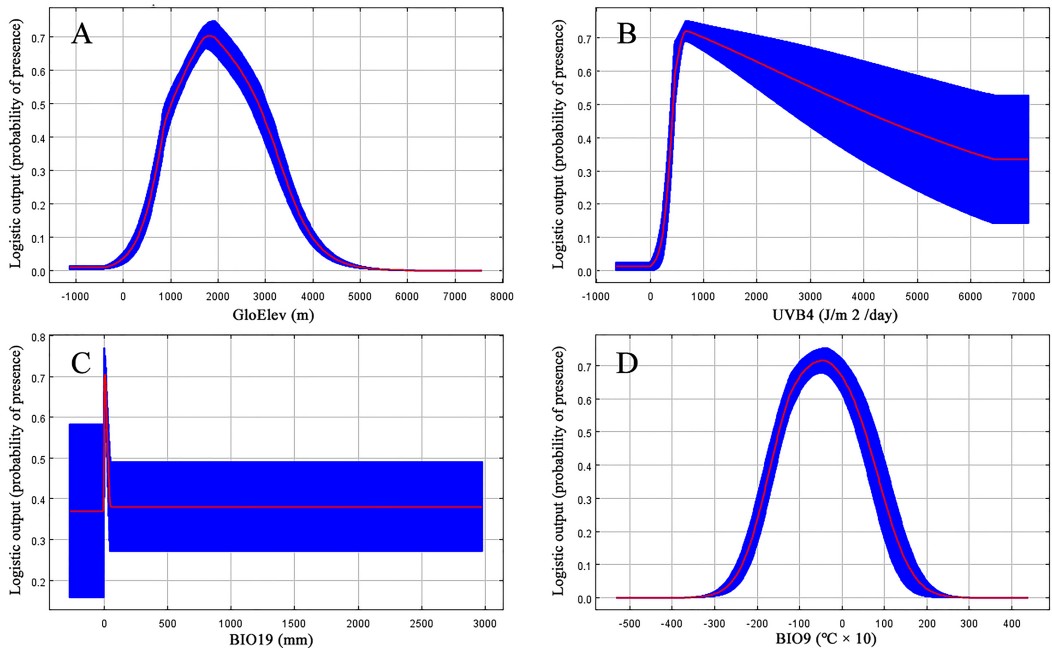

**Figure 4** Response curves for the four most important environmental predictors in the *Clematis* sect. *Fruticella* distribution model. (A) Elevation. (B) Mean UV-B of Lowest Month. (C) Precipitation of Coldest Quarter. (D) Mean Temperature of Driest Quarter.

Table 2). Based on high levels of greenhouse gas emissions (RCP 8.5), the estimated habitat in 2050 would increase mainly in the northern margin, but decrease in the western parts of the current suitable habitat (Fig. 5G, Table 2). Under the same pathway, Maxent estimated that by 2070 the range of suitable habitat would increase in both the southern and northern margins of the current suitable habitat and decrease mainly in west (Fig. 5H).

## Shift of the distribution center of suitable habitat

The current centroid of suitable habitat was estimated to be in Alxa League, Inner Mongolia (38°14′13″N, 105°13′24″E, Fig. 6). However, during the LIG period, the habitat centroid was in northern Ningxia province (40°42′46.8″N, 101°54′32.4″E). This centroid shifted to the southwest during the LGM (37°48′47″N, 105°3′22″E). In the MH, the centroid was very close to its current location (37°58′38″N, 104°49′35″E). Looking to a possible future under RCP 2.6, the centroid will have shifted eastward to central Ningxia Province (38°12′1″N, 106°22′53″E) by 2050, and then would move backward (38°22′23″N, 105°33′31″E) by 2070. Applying RCP 8.5, the centroid is predicted to shift northeast (38°47′19″N, 105°36′35″E) by 2050 and then shift eastward to Ordos, Inner Mongolia (38°40′4″N, 106°47′42″E) by 2070.

## DISCUSSION

The Loess Plateau of China, known by its arid and semi-arid climate, low vegetation coverage, erodible soil, and extensive monsoonal influence, has some of the most severe
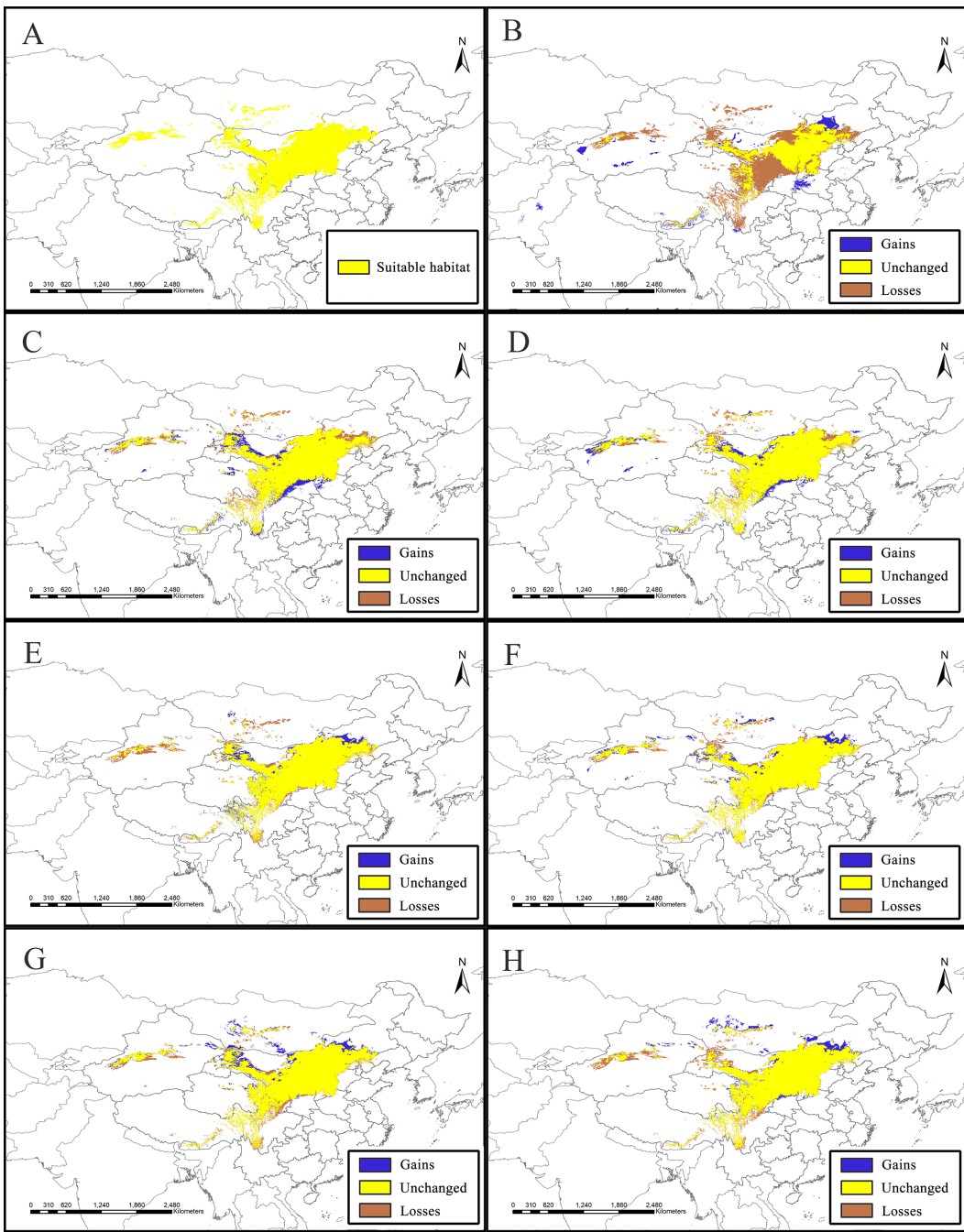

**Figure 5** **Species distribution models (SDMs) of *Clematis* sect. *Fruticella*.** Models B through H reflect habitat differences compared to the current model A. (A) Current. (B) Last Interglacial. (C) Last Glacial Maximum. (D) Mid Holocene. (E) Year of 2050 (under RCP2.6). (F) Year of 2070 (under RCP2.6). (G) Year of 2050 (under RCP8.5). (H) Year of 2070(under RCP8.5). RCP: representative concentration pathway.

**Table 2** Dynamic changes in the distribution area for *Clematis* sect. *Fruticella* under four future and three past climate scenarios compared to the current suitable habitat area.

| Scenarios | Increased (km²) | Unchanged (km²) | Decreased (km²) |
|---|---|---|---|
| Last interglacial | 153,522.601 | 660,109.5024 | 730,700.3814 |
| Last Glacial Maximum | 173,431.2386 | 121,6782.014 | 174,027.8697 |
| Mid Holocene | 147,854.6056 | 125,3035.203 | 137,774.6803 |
| [a]RCP2.6-2050 | 109,356.1992 | 121,7708.362 | 173,101.5214 |
| RCP2.6-2070 | 124,350.4807 | 125,8326.379 | 132,483.5045 |
| RCP8.5-2050 | 146,755.5483 | 124,6927.585 | 143,882.2986 |
| RCP8.5-2070 | 123,800.952 | 1,200,798.581 | 190,011.3027 |

**Notes.**
[a]RCP, representative concentration pathway.

soil and water loss conditions in the world (*Fang et al., 2011*; *Jiang et al., 2013*; *Wu et al., 2019*). This area's natural vegetation is a key factor controlling soil erosion and stopping desertification (*Zheng, 2006*; *Fang et al., 2011*). Species of *Clematis* sect. *Fruticella* members, important components of xerophytic vegetation in China (*He, Liu & Xie, 2018*), are found over the entire Chinese Loess Plateau and in adjacent arid areas (*Wang & Li, 2005a*). This study presented a detailed analysis of this native plant taxon's current suitable habitat, as well as its predicted habitat under past and future climate scenarios. Our goal was to provide important insights into the feasibility of using those species for soil fixation in the arid and semi-arid environments of northeast Asia.

Our models showed that *Clematis* sect. *Fruticella* could potentially be distributed in a wide range of arid or semi-arid areas in Mongolia and parts of Xinjiang, southeastern Xizang, and southern Sichuan provinces in northern and western China (Fig. 2D) where it is currently not found. Moderately and highly suitable habitats predicted by our model are consistent with this section's actual present distribution (725,110 km²). Because the section's predicted distribution area was broader than its actual distribution area, this study provides more expanded areas for the cultivation of this section's species.

When the nine tested variables were run through the jackknife test (Fig. 3), GloElev, UVB4, BIO19, and BIO9 were the dominant elements driving the section's potential and current distributions. The GloElev response curve (Fig. 4) showed that the estimated suitable elevation for this section is 539 to 3,620 m. This model estimation was highly consistent with, but a little wider, in range compared to the section's actual elevation. From specimen records, the most widely distributed species, *Clematis fruticosa*, occurs at 700 to 2,300 m. The western Loess Plateau species *C. nannophylla* has a wider elevation range, from 1,200 to 3,200 m. The most narrowly distributed species, *C. canescens*, also has a very narrow elevation range (1,200 to 1,500 m). The widely distributed species *C. tomentella* often occurs at 1,300 to 2,200 m. Finally, the north Hengduan Mountains species *C. viridis* has the highest elevation range, from 2,700 to 3,600 m. Therefore, the actual elevation distribution of the section is 700 to 3,600 m.

Non-ionizing solar UV-B radiation (280–315 nm) is an abiotic stress factor for sessile plant species (*Nawkar et al., 2013*; *Blagojevic et al., 2019*; *Derebe et al., 2019*). Too much

UV-B exposure damages both cell membranes and DNA and can cause morphological changes (*An et al., 2000*; *Schmitz & Weissenbock, 2003*; *Kataria & Guruprasad, 2012*). However, as a major energy source, plants need low-level UV-B radiation to help regulate growth and development (*Blunden & Arndt, 2012*). Since *Clematis* sect. *Fruticella* plants are heliophilous, it was no surprise that one of the UV-B factors, UV-B4, was one of the most important contributing factors in the model. It indicated that UV-B radiation must be at least 300.5 J/m$^2$/day for *Clematis* sect. *Fruticella* plants to thrive. Previous studies showed that long term UV-B exposure may result in smaller leaves and shorter internodes and plant heights (*Antonelli et al., 1997*; *Krizek, Mirecki & Britz, 2006*). Compared to other majority number of climbing and often compound-leafed *Clematis* species, the erect shrubby plants of *Clematis* sect. *Fruticella* also have very reduced simple leaves. Perhaps those characters developed as an adaptive response to strong sunlight.

A temperature (BIO9) and a precipitation (BIO19) variable in our model each contributed greatly to the predicted distribution of *Clematis* sect. *Fruticella*. (Table 1). They indicated that precipitation during the winter and temperatures during the dry season were important determinants of the section's distribution. Currently, the section's area of distribution is influenced greatly by a monsoonal climate (*Wang et al., 2019a*; *Wang et al., 2019b*), in which the coldest and driest season of the year is winter. Plants of *Clematis* sect. *Fruticella* are dormant in winter and their physiological activities are weak. However, a certain amount of precipitation and temperature accumulation in this period are required for dormancy breaking, vegetative bud development, and flower bud differentiation (*Xu et al., 2019*).

Changing climate has affected species distributions and received much attention in recent ecological studies, including those that have shown that some plant species have shifted their distributions drastically during Quaternary climate changes (*Lenoir et al., 2008*; *Qiu, Fu & Comes, 2011*). Distributions of many tree and shrub species typically showed latitudinal range shifts in response to glacial movements during the Quaternary (*Petit et al., 2003*; *Ikeda & Setoguchi, 2007*; *Qiu, Fu & Comes, 2011*). However, the estimated suitable habitat centroid of *Clematis* sect. *Fruticella* was very stable in our study. In our models, it shifted only slightly between the LIG era and the years 2050 and 2070, and no patterns of changing elevation were found in those centroid shifts (Fig. 6).

During the last glaciation, global temperature was 5–12 °C lower than they are now and the glacier areas were 8.4 times of the present time in China (*Wang & Liu, 2001*; *Li et al., 2004*; *Chen, Kang & Liu, 2011*). This greatly affected plant distributions. However, our model showed that suitable habitat for *Clematis* sect. *Fruticella* in the LGM did not shrink as much, as might be expected (Fig. 5, Table 2). On the contrary, the smallest area of suitable habitat was estimated to be in the LIG period (Fig. 5, Table 2), when it was only about half of the current area, but then expanded significantly in LGM. This situation is like those of other reported cold-adapted plant taxa, such as species of *Picea*, *Taxus*, *Tsuga*, and *Quercus* (*Li et al., 2013*; *Liu et al., 2013a*; *Liu et al., 2013b*; *Sun et al., 2015*; *Yu et al., 2015*; *Du et al., 2017*; *Zhang et al., 2018a*; *Zhang et al., 2018b*), in which low and more stable LGM temperatures created a dispersal opportunity for cold-adapted plant species (*Stewart et al., 2009*; *Kozhoridze et al., 2015*). *Clematis* is one of few cosmopolitan genera in

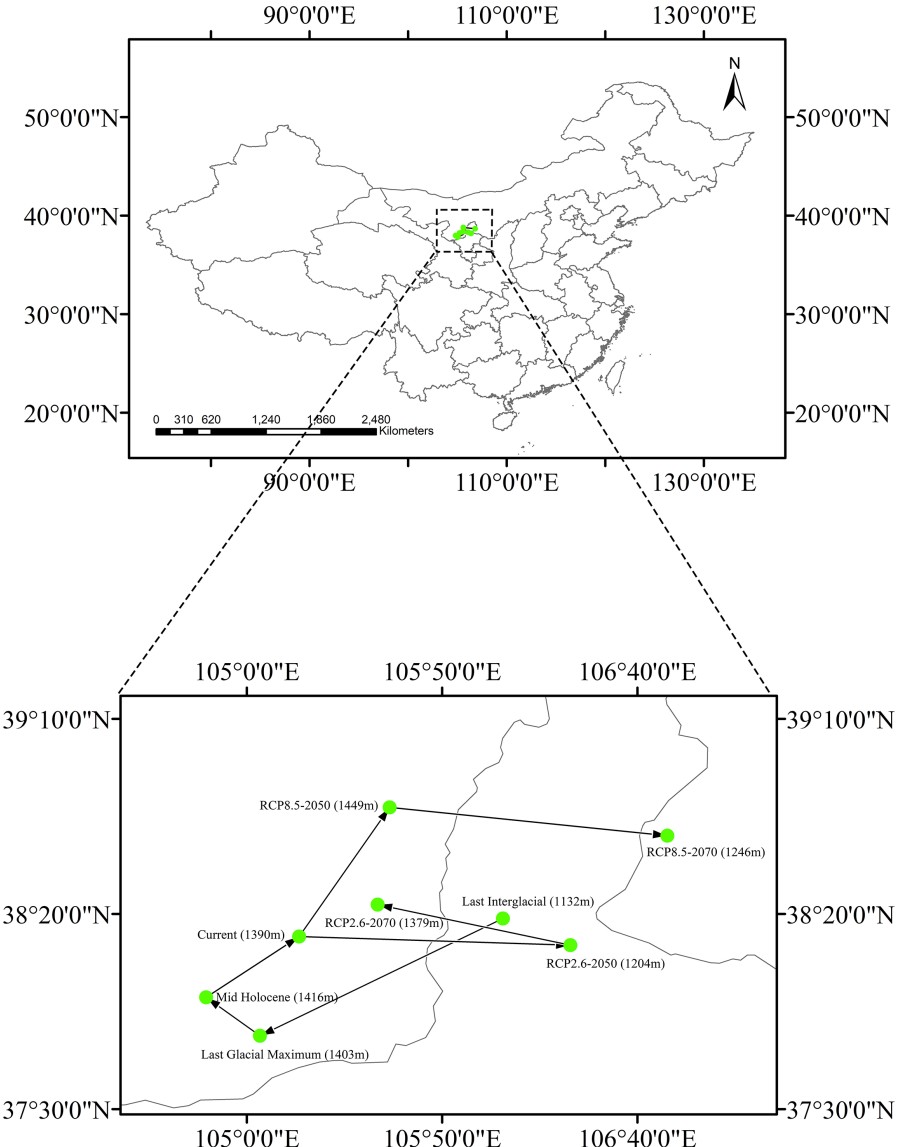

**Figure 6 Core distributional shifts of *Clematis* sect. *Fruticella* under the four future and three past climate scenarios.** RCP: representative concentration pathway.

the buttercup family, with some species found in subtropical to tropical areas. However, like most of the genera in Ranunculaceae, majority of *Clematis* species occur in temperate zones and are cold-adapted (e.g., sect. *Montana*, sect. *Atragene*, sect. *Meclatis*, and sect. *Fruticella*). Our models confirmed that the LGM's cold climate did not adversely affect *Clematis* sect. *Fruticella* distribution.

Study of the depositional sequences of paleosol and loess layers of the Loess Plateau showed that there were five strong summer and four winter monsoon events in the LIG period, demonstrating an unstable climate during that time (*Guan et al., 2007*). Paleoenvironmental reconstruction of the Loess Plateau implied that both the highest

temperature and precipitation amounts in the last 136,000 years occurred during the LIG period (18.1 °C, maximum mean annual temperature and 1,000 mm maximum mean annual precipitation) (*Lu et al., 2007*). That unstable and relatively warm, humid climate may have been the main reason that the suitable habitat for *Clematis* sect. *Fruticella* during the LIG period contracted to its smallest area.

In addition to the environmental factors discussed above, other factors, such as interspecific competition, may also influence the distribution of *Clematis* sect. *Fruticella*. During the LIG period, the Loess Plateau had mainly forest and forest-steppe vegetation (*Cai et al., 2013*). Such forest ecosystems would produce extensive shading, thus creating interspecific competition for solar radiation, a distinct minus for *Clematis* sect. *Fruticella*. Another study showed that *C. fruticosa*, now common and widely distributed in Loess Plateau, requires outcrossing, even though it is self-compatible (*Hou et al., 2016*). A forest environment could hinder pollen and fruit migration and prevent long-distance species dispersal. These ecological factors could contribute to the contraction of suitable habitat for *Clematis* sect. *Fruticella* during LIG.

Our model suggests that during the transition from the LIG to the LGM, the suitable habitat of *Clematis* sect. *Fruticella* expanded to an area very similar to what is seen today. Thus, the suitable habitat demonstrated stability from the LGM to the future scenarios. No significant area expansion/contraction or latitudinal/altitudinal shifts were detected after the LGM. As the climate changed from warm and humid (LIG) to cold and dry (LGM), the forest ecosystems were then changed into steppe and desert steppe predominantly occupied by *Artemisia* and Chenopodiaceae (*Cheng & Jiang, 2011*). During that time, the cold and drought tolerances of *Clematis* sect. *Fruticella* facilitated its adaption to the environment and fostered its expanded distribution. To the MH, the climate and vegetation of the Loess Plateau and adjacent areas were like those of the present (*Ge & Wei, 2008*; *Cheng & Jiang, 2011*), so the predicted suitable habitat of *Clematis* sect. *Fruticella* remains stable with only minor changes.

Distribution ranges of many plant species in the Northern Hemisphere may shift northward or to higher elevations in response to the global warming in the future (*Hof et al., 2011*; *Bai, Wei & Li, 2018*; *Wang et al., 2019a*; *Wang et al., 2019b*; *Zhang, Zhang & Tao, 2019*). However, *Clematis* sect. *Fruticella* would likely experience very limited changes in suitable habitat in the future. Although plants of this section are essentially cold adapted, they are also widely adaptable to warm temperatures during summer. Studies have indicated that the semi-arid regions in China will continue to expand, and less precipitation and soil moisture and increased drought frequency may occur because of global warming (*Chou et al., 2009*; *Huang et al., 2019*). These changes will not adversely affect the suitable habitat of the drought-resistant *Clematis* sect. *Fruticella*.

## CONCLUSIONS

Investigating the effects of climate change on both plant distribution and plant response mechanisms aids the development of effective conservation strategies and sustainable use of biodiversity. According to our modeling, the predicted current potential range of *Clematis*

sect. *Fruticella* was broader than its current distribution range, and the GloElev, UVB4, BIO19, and BIO9 variables most affected this section's distribution. Unlike many other plant taxa, *Clematis* sect. *Fruticella* underwent a significant range contraction during the LIG period, and then an expansion during the LGM to amounts similar to present amounts. Cold, dry, and relatively stable climate, as well as steppe or desert steppe environments may have facilitated the range expansion of this cold-adapted and drought-resistant plant taxon during the LGM. Although climate warming will likely intensify in the future, the suitable habitat of *Clematis* sect. *Fruticella* will not much change. Other factors, such as interspecific competition, human activities, and biological interactions (*Liu et al., 2016*) that could not be integrated into our modelling, may also affect the section's distribution. The results of our study help us understand distributional dynamics of *Clematis* sect. *Fruticella* and may aid the conservation and sustainable use of these important woody plants in Chinese arid and semiarid areas.

### Funding
This study was supported by the National Natural Science Foundation of China (no. 31670207 to Lei Xie), the Beijing Natural Science Foundation (no. 5182016 to Lei Xie), and the Medium- and Long-term Scientific Study Projects for Young Teachers of Beijing Forestry University (no. 2015ZCQ-BH-03 to Lei Xie). The funders had no role in study design, data collection and analysis, decision to publish, or preparation of the manuscript.

### Grant Disclosures
The following grant information was disclosed by the authors:
National Natural Science Foundation of China: 31670207.
Beijing Natural Science Foundation: 5182016.
Medium- and Long-term Scientific Study Projects: 2015ZCQ-BH-03.

### Competing Interests
The authors declare there are no competing interests.

### Author Contributions
- Mingyu Li and Jian He conceived and designed the experiments, performed the experiments, analyzed the data, prepared figures and/or tables, and approved the final draft.
- Zhe Zhao, Rudan Lyu and Min Yao performed the experiments, analyzed the data, prepared figures and/or tables, and approved the final draft.
- Jin Cheng and Lei Xie conceived and designed the experiments, authored or reviewed drafts of the paper, and approved the final draft.

### Data Availability
   The raw records of *Clematis* sect. *Fruticella* distribution sites are available in Table S1.

## Supplemental Information

Supplemental information for this article can be found online at http://dx.doi.org/10.7717/peerj.8729#supplemental-information.

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
