# Peer review of "Predictive modelling of the distribution of Clematis sect. Fruticella s. str. under climate change reveals a range expansion during the Last Glacial Maximum"

_PeerJ, doi:10.7717/peerj.8729_

## Round 0.1 · original submission · Major Revisions

You will find comments by the two reviewers below, I agree with them mainly regarding the methods. The need to perform additional ENMs for every species will improve forecasting distribution ranges of the species studied here. On the other hand detailed information on the use of the models is required as well. Statistics such like TSS, an independent measure to evaluate model performance will help the robustness of your models.

My suggestion is that in the Introduction you need to explain better why you selected this group in Clematis. Also, previous hypotheses of the influence of the climate of the LGM in distribution ranges of plant around the world and specifically in the area of study will help to understand the importance of climate variables in expanding or contracting distrubtion ranges. With this information it will be possible to propose hypotheses to be tested. The data on the Clematis species provided in Introduction is mostly taxonomic band ecological and pollination biology data will help understand the factors that might affect their distribution.

Reviewer 1 ·

Basic reporting

The manuscript titled “Predictive modelling of the distribution of Clematis
sect. Fruticella s. str. under climate change reveals a range expansion during the Last Glacial Maximum” coauthored by Mingyu Li, Jian He, Zhe Zhao, Rudan Lyu, Min Yao, Jin Cheng, Lei Xie is a well-written manuscript. The manuscript has good enough references and some of them relevant in the field of the study, and the aim of the manuscript is well presented.

Experimental design

The methods used to support the aims are good, but they can be better. For example, I encourage following the GBIF citation recommendation. The variables used for the analyses are good as well as the multicollinearity analysis among variables. The present and past scenarios are good. But the authors need to explain in detail why they are using the RCPs CCSM4 among many others that exist.

The critical part of the study is to combine the five Clematis species to do all the SDM analyses. I think every species must be modeled separately and then the sum of models, for the present, past, and future will be done. Because the ecological niche is a particular and unique for each species, not for section Fruticella.
The historical aspects of the models relating to species distribution (hypothetical ‘M’ region; Barve et al., 2011) must be used for every single species.
To evaluate the performance of the models I recommend calculating the true skill statistic (TSS), a threshold-dependent measure of model performance that evaluates the accuracy of predictive maps generated by presence-only data (Allouche et al., 2006; Liu et al., 2013).
The centroid of a suitable habitat 
must be evaluated for every single species.

Validity of the findings

Because the authors apply all the analyses to section Fruticella, not for every single species, it is possible that the results change and they need to discuss the new results.

Additional comments

I think your study is interesting for so many readers, but you need to apply the analyses for every single species and then discuss your findings.

Reviewer 2 ·

Basic reporting

I appreciate very much that you have thought of me for the review of the manuscript entitled “Predictive modelling of the distribution of Clematis sect. Fruticella s. str. under climate change reveals a range expansion during the Last Glacial Maximum”.

I have read the manuscript carefully, and I found if very interesting. The manuscript is written clearly and uses a good English language. The literature is useful and well referenced at the end of the manuscript. The figures show high quality and they are well labelled to help the reader. Most interesting contribution of this study is shown in the research about the effects of climate change on distribution and response mechanisms of plants. This paper addresses a relevant question regarding climate change context where organisms are displaying different responses to future shifts that will occur.

Experimental design

I would like to enhance that the methods sound very interesting for me, and I am mostly related with them. They are reasonable and generally well justified. It is appreciated that the authors use some novel approaches. I would like to value the inclusion of 35 environmental variables, not only 19 from WorldClim database, which is the most common option on these studies. I would also like to thank you for teaching me an unknown method to test the multicollinearity, in this case using ArcGIS software. Related to this Multicollinearity Analysis test among all variables (paragraph starts line 175) finally were kept the most ecologically meaningful. I would like to recommend to the authors the Variance Inflation Factor (VIF), a measure of how much collinearity increases variance in a model implemented in HH package (Heiberger, 2017) in R.
However, below I refer to some aspects which are unclear and require a justification in my opinion.

(1) I have a doubt relating the idea reflected in the introduction about those methods of species distribution modelling which use only presence data (Maxent) vs. others which use both presence and absence (GLM, GAM, and Random Forest). I think that the distribution of Clematis sect. Fruticella s. str. is well-known, such as is shown in the first paragraph of M&M (lines 130-149). In case that you cannot use real absence because you disown the accurate distribution of species in some region you could also use these algorithms implementing a binary response with presence and artificial absence data (pseudoabsence/background). I recommend you read the following article about this: “Selecting pseudo-absences for species distribution models: how, where and how many?” (Barbet-Massing et al., 2012 in Methods in Ecology and Evolution). If you use background you will have more points of this instead of presence. However, you could weigh given a determined value. I would like to compare the results obtained from Maxent with other algorithms. I would recommend you to use the package implemented in R “biomod2” which is used for ensemble platform for Species Distribution Modelling. Depends on your data, presences are considered as presences, absences as absences and pseudo-absences as background (recommendation is to use a large amount of random sampled pseudo-absences to be close to the background concept). With this package you could select the best model using Akaike Information Criteria (AIC). You can use the function “background_data_dir” from BIOMOD_ModellingOptions() or some random pseudo-absence sampling at BIOMOD_FormattingData() in case that you don’t feel reliability to obtain absence data as you said in the Introduction (line 65). If this is so, I suggest that you provide a more detailed explanation in the Methods paragraph. Thus, I would like to see the estimated results but including other species distribution modelling algorithms. I think that in this study could be very important to compare the potential suitable distribution areas predicted using not only Maxent modeling scenarios.

(2) In the Table 1 the authors show different resolution per each environmental variables. I would like to remember them the importance of use the same resolution for all variables used to predict a model within the same period of time (LGM, LIG, current, future).

Validity of the findings

I think that the results according to the model tested are very exciting for the study of the possible plant response mechanisms involved in the development of conservation of biodiversity under the effects of the current climate change. The results are very clear, although I would like to compare the same scenarios using other methodological algorithms in order to provide a more robust and reliable scientific contribution. It is very interesting to see how there are species which are suffering adaptations, whose distribution will not show drastic shifts in spite of the increasing of the global temperature. Conclusions are well linked to the original aim of this study.

Additional comments

I have read the manuscript carefully, and I found it very interesting. The manuscript is written clearly and uses an interesting and detailed bibliography with new and highly demanded approaches. However, to make this contribution more reliable I think that authors should carry out other methodological algorithms related to species distribution modeling.
I hope my suggestions and comments to the authors will contribute to improve the quality of the manuscript.
In any case I congratulate you for the excellent work!

---

## Round 0.2 · accepted · Accept

I appreciate your effort to consider all suggestions by the reviewers. Although the decision is to accept the paper, the only issue I found is that in Figure 5 the legend within every map says unchange and should say unchanged. So please have this figure corrected for the proofs that the editorial staff of PeerJ will send you soon.

Reviewer 1 ·

Basic reporting

In this new R1 version of the manuscript titled “Predictive modelling of the distribution of Clematis sect. Fruticella s. str. under climate change reveals a range expansion during the Last Glacial Maximum,” the authors took into account all our previous comments. The manuscript now is much better, is well-written and the aims of the manuscript are well presented.

Experimental design

The authors did a good justification for my previous comments. Additionally, they applied the TSS test to validate their models.

Validity of the findings

The authors justify pretty well why they did the SDM analyses to section Fruticella instead of a single species analysis.

Additional comments

The new version is pretty good and I want to acknowledge to the authors for attending all our previous comments in this new version.